# Diagnosis of Low-Grade Urothelial Neoplasm in the Era of the Second Edition of the Paris System for Reporting Urinary Cytology

**DOI:** 10.3390/diagnostics13162625

**Published:** 2023-08-08

**Authors:** Konstantinos Christofidis, Napoleon Moulavasilis, Evangelos Fragkiadis, Dimitrios Goutas, Andreas C. Lazaris, Dionisios Mitropoulos, Panagiota Mikou

**Affiliations:** 1Cytopathology Department, Laiko General Hospital, 11527 Athens, Greece; konstantinos.christofidis@gmail.com; 21st Urology Department, National and Kapodistrian University of Athens, Laiko General Hospital, 11527 Athens, Greece; napomoul@hotmail.com (N.M.); e.fragkiadis@gmail.com (E.F.); ourologiki@laiko.gr (D.M.); 31st Laboratory of Pathology, National and Kapodistrian University of Athens, Laiko General Hospital, 11527 Athens, Greece; goutas.dimitris@hotmail.com (D.G.); alazaris@med.uoa.gr (A.C.L.)

**Keywords:** cyto-histopathological correlation, low-grade urothelial neoplasm (LGUN), non-high-grade urothelial carcinoma (NHGUC), TPS, urinary cytology

## Abstract

Background: The Paris System for Reporting Urinary Cytology (TPS) is considered the gold standard when it comes to diagnostic classifications of urine specimens. Its second edition brought some important changes, including the abolition of the diagnostic category of “low-grade urothelial neoplasm (LGUN)”, acknowledging the inability of cytology to reliably discern low-grade urothelial lesions. Methods: In this retrospective study, we assessed the validity of this change, studying the cytological diagnoses of histologically diagnosed low-grade urothelial carcinomas during a three-year period. Moreover, we correlated the sum of the urinary cytology diagnoses of this period with the histological diagnoses, whenever available. Results: Although all the cytological diagnoses of LGUN were concordant with the histological diagnoses, most low-grade urothelial carcinomas were misdiagnosed cytologically. Subsequently, the positive predictive value (PPV) of urinary cytology for the diagnosis of LGUN was 100%, while the sensitivity was only 21.7%. Following the cyto-histopathological correlation of the sum of the urinary cytology cases, the sensitivity of urinary cytology for the diagnosis of high-grade urothelial carcinoma (HGUC) was demonstrated to be 90.1%, the specificity 70.8%, the positive predictive value (PPV) 60.3%, the negative predictive value (NPV) 93.6% and the overall accuracy 77.2%, while for LGUN, the values were 21.7%, 97.2%, 87.5%, 58.6% and 61.9%, respectively. Risk of high-grade malignancy was 0% for the non-diagnostic (ND), 4.8% for the non-high-grade urothelial carcinoma (NHGUC), 33.3% for the atypical urothelial cells (AUCs), 65% for the suspicious for high-grade urothelial carcinoma (SHGUC), 100% for the HGUC and 12.5% for the LGUN diagnostic categories. Conclusions: This study validates the incorporation of the LGUN in the NHGUC diagnostic category in the second edition of TPS. Moreover, it proves the ability of urinary cytology to safely diagnose HGUC and stresses the pivotal role of its diagnosis.

## 1. Introduction

The role of urinary cytology as a diagnostic test for the detection and surveillance of urothelial cancer is crucial. Bladder cancer represents the tenth (10th) most common form of cancer worldwide, the fourth (4th) most common in men and the seventeenth (17th) most common in women [1]. It shows remarkably high recurrence and progression rates, constituting, therefore, a significant burden on healthcare systems [2]. Cystoscopy and urinary cytology play a significant role in the diagnosis of patients who present with urinary tract symptomatology but also in the follow-up of patients who have already received treatment for their bladder cancer. Nevertheless, cystoscopy is an invasive, painful and costly procedure, with notable false-positive and false-negative results. Urinary cytology is characterized by a high sensitivity for the detection of high-grade tumors and in situ carcinomas (CISs), but a low sensitivity for low-grade tumors.

Additionally, until recently, urinary cytology lacked a consistent and valid grading system, leading to significant interobserver and interinstitutional variability of diagnoses. More specifically, the biggest problem was the significant number of indeterminate diagnoses, such as “atypical cells”, which are of limited clinical utility. Urinary cytology can be quite challenging, in part due to the difficulty of identifying low-grade urothelial neoplasms (LGUNs), but also because of technical problems, such as cellular degeneration prior to fixation and inadequate sample cellularity. These issues created the necessity for the creation of a standardized and comprehensive reporting system for urinary cytology that would be able to decrease the frequency of indeterminate diagnoses, create diagnostic categories correlated with the risk of high-grade malignancy (ROHM) and increase the overall sensitivity of urinary cytology.

In 2016, the Paris System for Reporting Urinary Cytology was introduced by the International Academy of Cytology and the American Society of Cytopathology. This reporting system presented clearly defined criteria for each of the six diagnostic categories: inadequate specimen (inadequate/nondiagnostic), negative for high-grade urothelial carcinoma (NHGUC), atypical urothelial cells (AUCs), suspicious for high-grade urothelial carcinoma (SHGUC), high-grade urothelial carcinoma (HGUC) and low-grade urothelial neoplasia (LGUN).

The introduction of the Paris System for Reporting Urinary Cytology was a significant evolution for urinary cytology [3,4]. Decades after the first implementation of urinary cytology in the diagnosis of urinary tract pathology [5], the aim of the examination was shifted to the diagnosis of high-grade urothelial carcinoma [6]. This has contributed to the formation of clearer and robust diagnostic criteria and ameliorated the correlation with the risk of high-grade malignancy (ROHM) of each diagnostic category. In addition, it has changed the pathologists’ diagnostic perception and has improved communication with clinicians [7,8,9,10]. TPS has also been applied in specific settings, like upper urinary tract specimens and BCG-treated patients [11,12,13,14], as well as in specimens with different preparation techniques [15]. Reviews of TPS report a decrease in the frequency of intermediate diagnoses and an acceptable interobserver variability [16,17,18,19,20].

A separate category of “low-grade urothelial neoplasm” with its own diagnostic criteria was included in the first edition of TPS. In the second edition, this category was abolished and LGUN was incorporated in the “negative for high-grade urothelial carcinoma” diagnostic category [21]. This change is justified by past and recent studies in the literature [22,23]. However, there are a few institutions that still share a rather optimistic view about the competence of cytology in diagnosing LGUN and are thus opposed to this alteration. In our department, having dealt with urinary cytology specimens throughout the years, we adopted and have been using TPS since 2016. Nevertheless, we were skeptical about the aforementioned change. Thus, we searched our institution’s database and planned a study in order to investigate the validity of the incorporation of the “low-grade urothelial neoplasm” in the “negative for high-grade urothelial carcinoma” diagnostic category in the second edition of TPS, while also studying the accuracy parameters of the HGUC diagnosis.

## 2. Materials and Methods

### 2.1. Patients’ Selection

Our hospital’s histopathology database was searched for all the urinary specimens that were diagnosed as low-grade urothelial carcinomas (LGUCs) according to the 2016 WHO classification [24] between the years 2019 and 2021. This was followed by a search of our laboratory’s database for urinary cytology samples, obtained from the same patients during a period of one month before the biopsy. Cases with a cytological diagnosis of LGUN were reviewed by one cytologist and the main diagnostic features were noted. Additionally, we searched the cytopathology database for all cytology urinary samples that were diagnosed during this period. Thus, we compiled two distinct datasets for the period of 2019–2021 that were used to assess the accuracy parameters of the diagnosis of HGUC and LGUN.
Dataset 1: All histologically diagnosed LGUC cases for which a prior urinary cytology examination was performed.Dataset 2: All urinary cytology cases for which subsequent histopathological diagnoses were available.

The study material included voided urine, bladder or ureteral washings and catheterized urine samples. All relevant clinical information and data for each case were noted.

Consent for any retrospective study was given by all the patients included in the study and our hospital’s ethical board committee approved this study. This study was designed and conducted in accordance with the ICH protocol (Harmonized Tripartite Guidelines for Good Clinical Practice) and is harmonized with the principles of the Declaration of Helsinki.

### 2.2. Specimen Processing

All urinary samples that are received in our department are handled by specialized cytotechnologists. Slides are prepared both by the conventional method (Cytospin 3, Shandon Scientific Ltd., Cheshire, UK) and by liquid-based cytology (Thinprep 2000 system; Cytyc Co., Boxborough, MA, USA) in a split-sample manner. Every urinary sample is centrifuged at 2000 rpm for 10 min, the supernatant is discarded, and the sediment is divided in two parts. One part is centrifuged in the Cytospin 3 at 2000 rpm for ten minutes to prepare two slides, while the other one is stored in a PreservCyt solution. In cases with three urinary samples of the same patient from three consecutive days, only one Thinprep slide is prepared, using material from all three samples. In cases with only one sample, two cytospin slides and one Thinprep slide are prepared. In all cases, out of the two cytospin slides, one is ethanol-fixed and stained with the Papanicolaou stain and the other is air-dried and stained with the Giemsa stain, while the Thinprep slide is stained only with the Papanicolaou stain. The specialized cytopathologists of our department diagnose all cases according to the Paris System for Reporting Urinary Cytology (Table 1), and all difficult cases, as well as cases suspicious or positive for malignancy, are reviewed by a second cytopathologist.

### 2.3. Data Analysis

From dataset 1, the sensitivity and the positive predictive value of the diagnosis of LGUN were estimated. From dataset 2, we estimated the sensitivity, specificity, positive predictive value, negative predictive value, and overall accuracy of the diagnoses of LGUN and HGUC. Additionally, the risk of high-grade malignancy was assessed for all TPS categories in dataset 2.

## 3. Results

During the three-year period of 2019–2021, 487 urinary specimens were diagnosed as low-grade urothelial carcinomas in our hospital’s histopathology department and a precedent cytology report was available for 129 of these cases. For the latter, the male-to-female ratio was 2.5 and the mean patient age was 72 years. In this study, 23 out of the 129 patients included had a prior history of urothelial neoplasm and were under surveillance. Five of the cases were upper urinary tract washings (UUTs).

The corresponding cytology diagnoses for the included cases were as follows: 5 nondiagnostic/inadequate (ND), 52 negative for high-grade urothelial carcinoma (NHGUC), 30 atypical urothelial cells (AUCs), 14 suspicious for high-grade urothelial carcinoma (SHGUC) and 28 low-grade urothelial neoplasm (LGUN). There was no case with a high-grade urothelial carcinoma (HGUC) diagnosis (Table 2 and Figure 1). Based on these cases, the sensitivity of the cytological diagnosis of LGUN is 21.7%, while the positive predictive Value (PPV) is 100% (Table 3).

During the same period, a total of 1352 urinary samples were received in our cytopathology laboratory. Corresponding histopathologic reports were retrieved for 276 of these cases. For the latter, the male-to-female ratio was 1.8 and the mean patient age was 69 years. In total, 45 out of these 276 patients had a prior history of urothelial neoplasm and were under surveillance. Twelve cases were upper urinary tract washings (UUTs). The total number of cases in the years under study is lower than our departmental average due to the COVID-19 pandemic, which seriously affected our laboratory’s workload.

The cytological diagnoses for these cases were as follows: 5 ND, 103 NHGUC, 60 AUC, 40 SHGUC, 36 HGUC and 32 LGUN (Figure 2). The cyto-histopathological correlation is presented in Table 4. Accuracy parameters regarding the diagnosis of HGUC and LGUN were estimated separately. For HGUC, the cytological diagnoses of ND, NHGUC and LGUN were considered negative, while AUC, SHGUC and HGUC were considered positive. The sensitivity was 90.1%, the specificity 70.8%, the PPV 60.3%, the NPV 93.6% and the overall accuracy 77.2% (Table 5). ROHM was 0% for ND, 4.8% for NHGUC, 33.3% for AUC, 65% for SHGUC, 100% for HGUC and 12.5% for LGUN (Table 6).

For LGUN, the accuracy of cytology was assessed considering only LGUC as positive and all the other diagnostic categories as negative. The sensitivity was 21.7%, the specificity 97.2%, the PPV 87.5%, the NPV 58.6% and the overall accuracy 61.9% (Table 7).

A slide review was performed for the 28 cases with a cytological diagnosis of LGUN. Papillary urothelial tissue fragments (UTFs) with fibrovascular cores were present in 5 of the cases (one of them from the upper urinary tract) (Figure 3). Urothelial tissue fragments without conspicuous fibrovascular cores were noted in another 9 cases (Figure 4), while the remaining 14 cases showed marked cellularity with dispersed single tumor cells characterized by mild nuclear enlargement, lack of hyperchromasia, mild nuclear contour irregularities and elongated cytoplasm (cercariform cells) (Figure 5 and Figure 6). We did not review cases other than the ones with an LGUN diagnosis, as it would be very unlikely for them to present features characteristic of LGUN.

## 4. Discussion

The WHO Classification of Urinary and Male Genital Tumors (5th edition) classifies urothelial neoplasms according to the level of architectural disorder and cytological abnormality. This approach has not only proven to be clinically relevant but is also in accordance with the two major molecular pathways that play a role in the pathogenesis of urothelial neoplasms. The Paris System for Reporting Urinary Cytology is also included in the most recent WHO Classification [25].

Noninvasive papillary urothelial neoplasms are diagnosed as urothelial papilloma, papillary urothelial neoplasm of low malignant potential, noninvasive papillary urothelial carcinoma, low-grade, and high-grade. Papillary urothelial hyperplasia and urothelial proliferation of undetermined malignant potential are no longer recognized as separate entities, but as a part of the aforementioned neoplasms. On the other hand, the only diagnostic category recognized for noninvasive flat urothelial neoplasms is urothelial carcinoma in situ. Urothelial dysplasia, describing a flat lesion that lacks the appropriate criteria for a diagnosis of carcinoma in situ, is also not considered a separate diagnostic entity.

TPS has moved the focus of urinary cytology from the diagnosis of malignancy to the diagnosis of HGUC. The abolition of the sixth diagnostic category, the LGUN category, in the second edition of TPS represents a further step in this direction. However, there is still a debate about the justification of this decision. Cytology departments with long experience in exfoliative urinary cytology believe that this change is not reflecting the actual potential of urinary cytology for diagnosing LGUN. In this context, we have tried to assess our laboratory’s performance in this field.

We have conducted a study of LGUN cytology collecting data in two ways: (a) we identified all the histologically diagnosed low-grade urothelial carcinomas and correlated them with precedent cytology reports, and (b) we searched the cytology database for all urinary cytology cases and subsequently correlated them with the histopathology results, whenever available. We followed this two-way approach in order to thoroughly assess the performance of urinary cytology. Still, a considerable number of patients with a histological diagnosis of LGUC or HGUC have not previously gone through a cytologic examination. This is because urologists do not send samples for a cytologic examination from all patients with a suspected urinary tract neoplasm. Patients with symptoms like hematuria and/or radiographic evidence of a tumor may be directly subjected to a cystoscopy. There is also an important percentage of patients with only a cytological diagnosis, not followed by cystoscopy. Moreover, some of the patients may have left the hospital without any follow-up data available.

In our first dataset, 28 out of the 129 (21.7%) low-grade urothelial carcinomas were correctly diagnosed by cytology as LGUNs, showing a 100% PPV of urinary cytology. There were no false-positive cases, meaning no low-grade urothelial carcinomas were misdiagnosed as high-grade. However, 52 cases (40.3%) were diagnosed as negative (NHGUC). This high percentage of false-negative results has also been reported by other researchers [20]. Most of the histologically diagnosed LGUCs were reported as NHGUC by cytology in the studies reviewed. Thirty cases (23.2%) showed atypical urothelial cells (AUCs). Another 14 cases (10.8%) were diagnosed as suspicious for high-grade urothelial carcinoma (SHGUC).

In the second dataset, 5 cases were ND, 103 were diagnosed as NHGUC, in 60 cases atypical cells were found (AUC), 40 cases were SHGUC, while 36 were diagnosed as HGUC and 32 as LGUN. In total, 28 of the latter had a subsequent histologic confirmation (LGUC), while the remaining 4 cases were false positives with histology diagnosing HGUC. Moreover, 101 were false negatives, being diagnosed as ND (5 cases), NHGUC (52 cases), AUC (30 cases) and SHGUC (14 cases), while histologically diagnosed as LGUC.

The ability of cytology to diagnose LGUN was questioned long before the application of TPS [25,26]. Our results show that a diagnosis of LGUN in cytology usually corresponds to low-grade urothelial carcinoma in histology, with PPVs of 100% and 87.5% in our two different datasets. Furthermore, the specificity of the cytological diagnosis of LGUN—estimated in the second database only—was 97.2%. This probably explains our optimistic perception of cytology’s ability to diagnose LGUN and is in accordance with an earlier study by Raab et al., who declared high sensitivity and specificity for the cytological diagnosis of LGUN [27].

On the contrary, the sensitivity of cytology in diagnosing LGUN, estimated in both datasets, was only 21.7%, which is very low and reflects cytology’s inability to safely discern LGUN. In the literature, the sensitivity of cytology in that regard has been reported to be 10–70% [20,23,28,29]. Additionally, the NPV of the cytological diagnosis of LGUN was only 58.6% due to the large number of false-negative results, the majority of them being NHGUC, AUC and SHGUC.

Nuclear enlargement and hyperchromasia are not considered morphological features suggestive of LGUN, as researchers have pointed out in the past [27]. In the era of TPS, urothelial tissue fragments are considered the most important finding, consistent with LGUN histology. Papillary UTFs with fibrovascular cores were present in 5 out of 28 cases (17.8%), while UTFs without a papillary configuration and fibrovascular cores were present in 9 cases (32.1%). Although these features have been reported from the early days of exfoliative urinary cytology [5] and are considered the hallmark of LGUN by TPS [4], they are unfortunately not consistently present. Noteworthily, their presence is also reported in benign conditions, like urolithiasis [30]. Fourteen of the cases showed a single-cell pattern with marked cellularity, subtle nuclear atypia and cercariform cells. Murata et al. have described two cell patterns for LGUN cytology, the “isolated cell pattern” and the “cluster pattern” [31]. The single-cell pattern is not described as a consistent LGUN finding in TPS. It seems that although we have applied TPS since 2016, we have made LGUN diagnoses based on other, less well-defined criteria, like hypercellularity and presence of cercariform cells. This represents a limitation in our study since our laboratory’s sensitivity for LGUN would potentially have been even lower if TPS criteria were strictly applied in all cases.

The majority of LGUC cases (78.3%) were misdiagnosed by cytology. The percentage of cases diagnosed as negative was 40.3%, which is in keeping with the literature data [22,23,26], thus highlighting the subtle morphological changes of cells in LGUC. Interestingly, Zhang et al. have emphasized the variation in LGUN cytomorphology according to specimen type [32]. Another 23.2% of the histologically diagnosed LGUC were diagnosed as AUC, while 10.8% were considered as SHGUC. It is obvious that the clearly defined TPS criteria are not always easily implemented.

Although the main purpose of our study was the ability of cytology to safely diagnose LGUN, we evaluated its performance regarding the diagnosis of HGUC as well. Sensitivity was estimated to be 90.1%, specificity 70.8%, PPV 60.3%, NPV 93.6% and overall accuracy 77.2%. These results are in accordance with our previously published data [33] and other reports in the literature [10,20,34]. This proves the reliability of cytology in diagnosing HGUC.

The risk of high-grade malignancy was 0% for ND, 4.8% for NHGUC, 33.3% for AUC, 65% for SHGUC, 100% for HGUC and 12.5% for LGUN. Pastorello et al. reviewed the literature and reported ROHM ranging from 8.7% to 36.8% for NHGUC, 12.3% to 60.9% for AUC, 33.3% to 100% for SHGUC, and 58.8% to 100% for HGUC. Each institution provides different data; thus, it is very important for cytopathology labs practicing urinary cytology to assess their own data. Clinicians can benefit from this information by obtaining better comprehension of the cytological results and their significance, so that patient management, including therapeutic decisions and surveillance strategies, can be planned in the most appropriate manner.

Our study bears the limitations of its retrospective nature. Another limitation is the relatively low number of cases, attributed to the COVID-19 pandemic, which substantially decreased the annual number of specimens received by our laboratory in 2020 and 2021 compared to previous years.

## 5. Conclusions

Our conclusion is that the decision of the new edition of TPS to no longer recognize LGUN as a separate diagnostic category is justified. There is no way to consistently diagnose LGUN correctly by urinary cytology. Its incorporation in the NHGUC category, with a special note pointing to the possibility of it pertaining to a LGUN, reflects an approach characterized by diagnostic pragmatism. Fortunately, most of these tumors are diagnosed by radiology and cystoscopy, followed by a histopathologic examination of the cystoscopically biopsied lesion. Consequently, focusing on the diagnosis of HGUC is a clear, valid, and clinically relevant approach and should consequently be the aim of urinary cytology.

## Figures and Tables

**Figure 1 diagnostics-13-02625-f001:**
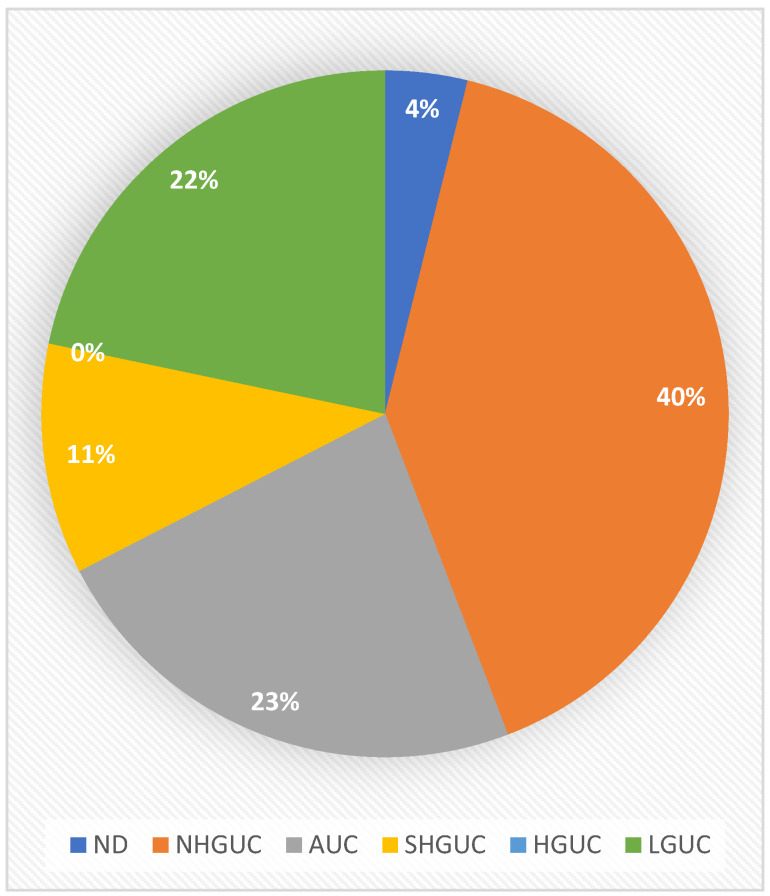
Distribution of all histologically diagnosed low-grade urothelial carcinomas in TPS categories (dataset 1).

**Figure 2 diagnostics-13-02625-f002:**
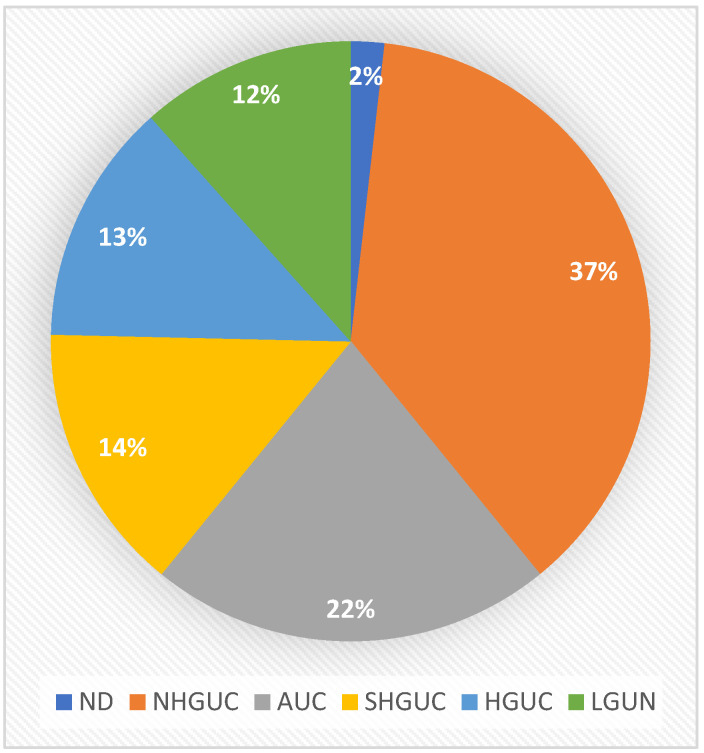
Distribution of all cytology cases in TPS categories (dataset 2).

**Figure 3 diagnostics-13-02625-f003:**
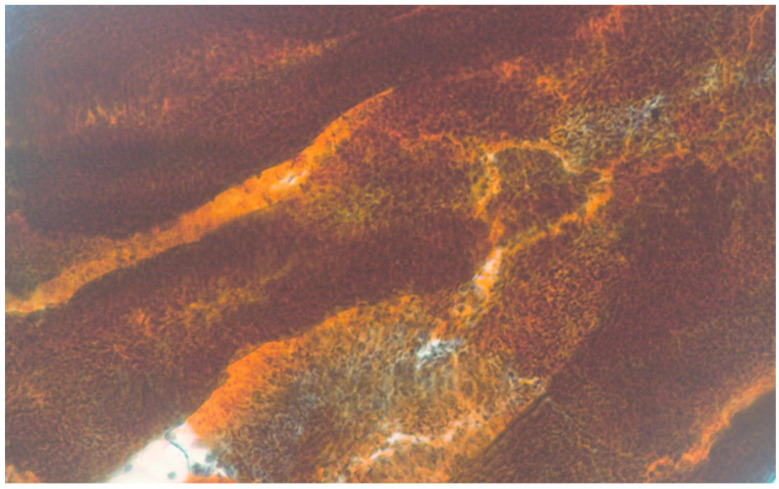
Papillary urothelial tissue fragment with fibrovascular cores from a ureteral washing. Pap stain, ×100.

**Figure 4 diagnostics-13-02625-f004:**
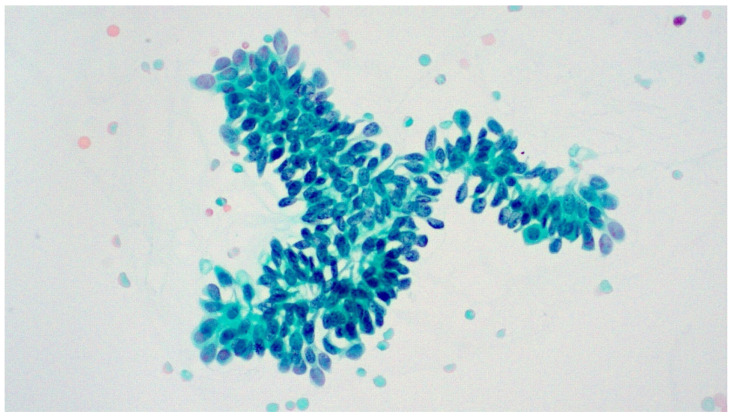
Papillary urothelial tissue fragment (fibrovascular core?) from voided urine. Pap stain, ×200.

**Figure 5 diagnostics-13-02625-f005:**
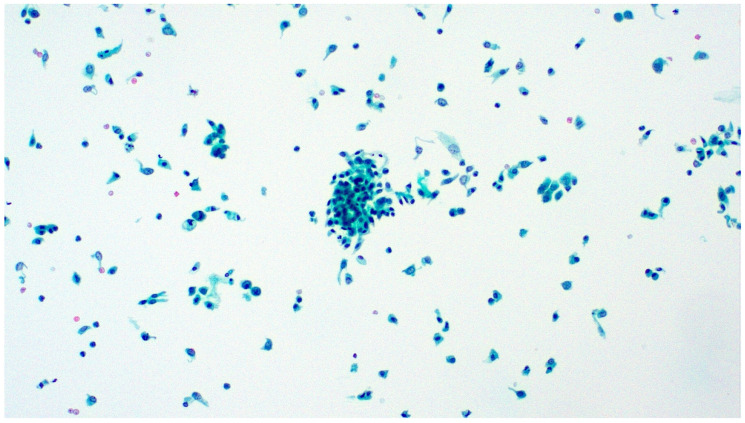
Urothelial tissue fragment and dispersed, mildly atypical cercariform cells from voided urine. Pap stain, ×200.

**Figure 6 diagnostics-13-02625-f006:**
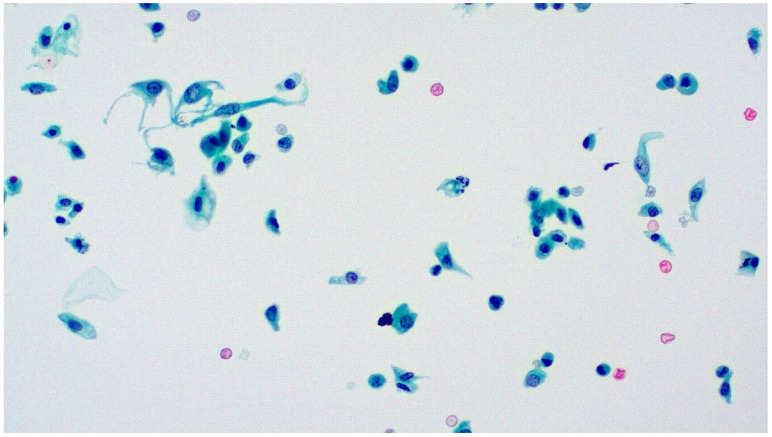
Single cell pattern. Dispersed, mildly atypical cercariform cells. Pap stain, ×200.

**Table 1 diagnostics-13-02625-t001:** The Paris System for Reporting Urinary Cytology (TPS).

1st Edition		2nd Edition	
1	Nondiagnostic/Unsatisfactory **(ND)**	1	Nondiagnostic **(ND)**
2	Negative for High-Grade Urothelial Carcinoma **(NHGUC)**	2	Negative for High-Grade Urothelial Carcinoma **(NHGUC)**
3	Atypical Urothelial Cells **(AUC)**	3	Atypical Urothelial Cells **(AUC)**
4	Suspicious for High-Grade Urothelial Carcinoma **(SHGUC)**	4	Suspicious for High-Grade Urothelial Carcinoma **(SHGUC)**
5	High-Grade Urothelial Carcinoma **(HGUC)**	5	High-Grade Urothelial Carcinoma **(HGUC)**
6	Low-Grade Urothelial Neoplasm **(LGUN)**		-
	Other		Other

**Table 2 diagnostics-13-02625-t002:** Cyto-histopathological correlation of low-grade urothelial carcinomas (dataset 1).

Cytology	ND	NHGUC	AUC	SHGUC	HGUC	LGUN	Total
Histology							
**LGUC (N)**	5	52	30	14	-	28	129
%	3.9	40.3	23.3	10.8	0	21.7	100

**Table 3 diagnostics-13-02625-t003:** LGUN: Sensitivity and PPV of urinary cytology (dataset 1).

Accuracy Parameters	%
Sensitivity	21.7
Positive Predictive Value	100

**Table 4 diagnostics-13-02625-t004:** Cyto-Histopathological correlation of all cytology cases (dataset 2).

Histology	Negative	LGUC	HGUC	Other	Total
Cytology					
**ND**	-	5	-	-	5
**NHGUC**	45	52	5	1	103
**AUC**	10	30	20	-	60
**SHGUC**	-	14	26	-	40
**HGUC**	-	-	36	-	36
**LGUN**	-	28	4	-	32
Total	55	129	91	1	276

**Table 5 diagnostics-13-02625-t005:** Accuracy parameters for the cytological diagnosis of HGUC (dataset 2).

Accuracy Parameters	%
Accuracy	77.2
Sensitivity	90.1
Specificity	70.8
Positive Predictive Value	60.3
Negative Predictive Value	93.6

**Table 6 diagnostics-13-02625-t006:** Risk of high-grade malignancy (ROHM) in TPS categories (dataset 2).

TPS	ND	NHGUC	AUC	SHGUC	HGUC	LGUN
**ROHM**	0	4.8%	33.3%	65%	100%	12.5%

**Table 7 diagnostics-13-02625-t007:** Accuracy parameters for the cytological diagnosis of LGUN (dataset 2).

Accuracy Parameters	%
Accuracy	61.9
Sensitivity	21.7
Specificity	97.2
Positive Predictive Value	87.5
Negative Predictive Value	58.6

## Data Availability

The data presented in this study are available in article.

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
