# Peer review of "Diagnosis of Low-Grade Urothelial Neoplasm in the Era of the Second Edition of the Paris System for Reporting Urinary Cytology"

_diagnostics, 2023, doi:10.3390/diagnostics13162625_

Round 1

Reviewer 1 Report

Dear authors, your study is clear, well presented, and results support the newest modifications in classification of the urine cytology. I have enjoyed reading it.

I have a few minor suggestions for corrections:

In Table 1: column two, , category six: it should be written "Low-Grade Urothelial Neoplasm" instead "Low-Grade Urothelial Carcinoma".

Chart 1: may be unnecessary.

Figure 3: it is blurred, please try to obtain a sharper photograph.

Discussion: please comment more about the high PPV of cytology in LGUN, because this is also important result.

English is fine, just a few small mistakes to correct.

Author Response

Greetings.

Thank you very much for your feedback. I hope this updated version covers all the issues you mentioned.

Best regards

Reviewer 2 Report

Dear Authors,

I read with interest your manuscript entitles: Diagnosis of Low-Grade Urothelial Neoplasm in the era of the 2nd edition of the Paris System for Reporting Urinary Cytology

Although the topic is very interesting and fascinating, while I appreciate the effort of the authors, I believe that this topic is outdated. 

The low-grade urinary cytology of the previous classification had limitations too great to overcome, primarily the low sensitivity. Therefore, the work, certainly supportive of what already exists in the literature, lacks the character of originality. 

I would advise the authors to take advantage of such data to complete a new study.

English should be revised in part just for the form.

Author Response

Greetings.

Thank you very much for your feedback. We understand that the discussion and research around LGUN cytology has been going on for some time. Nevertheless, considering the recent publication of the 2nd edition of The Paris System and the changes it brought, it is our belief that our study can play a role in supporting the value and general acceptance of TPS.

However, based on your comments we tried to implement more information and analyses in the study. We sincerely hope this updated version of our article will cover some of the issues you pointed out.

Best regards

Reviewer 3 Report

It was a very interesting study and within the scope of the journal.   Level of Originality, Writing Style and Clarity is good in this article. The idea for the main prescription is very good and the design for the study is acceptable.

1)      There are few grammatical errors which needs proofreading.

2)      The Keywords should be checked in Mesh database.

3)      The keywords should be sorted alphabetically.

4)      Used short and simple sentences.

5)      Authors did not comment on the gap of knowledge the current research will try to fill.

6)      Report the ethical code.

7)      The discussion section needs to be more elaborative.

8)      Please expand conclusion part.

There are few grammatical errors which needs proofreading.

Author Response

Greetings.

Thank you very much for your feedback. I hope this updated version of the article covers all the issues you mentioned.

Best regards

Round 2

Reviewer 2 Report

Dear authors, 

by changing the setting of the paper and having projected it in this new direction I believe that it is sufficient for publication.